# Sex-Specific Whole-Transcriptome Analysis in the Cerebral Cortex of FAE Offspring

**DOI:** 10.3390/cells12020328

**Published:** 2023-01-15

**Authors:** Nitish K. Mishra, Pulastya Shrinath, Radhakrishna Rao, Pradeep K. Shukla

**Affiliations:** 1Department of Structural Biology, St. Jude Children’s Research Hospital, Memphis, TN 38105, USA; 2Department of Physiology, University of Tennessee Health Science Center, Memphis, TN 38163, USA

**Keywords:** FAE, brain, RNA-seq, sex-specific, offspring

## Abstract

Fetal alcohol spectrum disorders (FASDs) are associated with systemic inflammation and neurodevelopmental abnormalities. Several candidate genes were found to be associated with fetal alcohol exposure (FAE)-associated behaviors, but a sex-specific complete transcriptomic analysis was not performed at the adult stage. Recent studies have shown that they are regulated at the developmental stage. However, the sex-specific role of RNA in FAE offspring brain development and function has not been studied yet. Here, we carried out the first systematic RNA profiling by utilizing a high-throughput transcriptomic (RNA-seq) approach in response to FAE in the brain cortex of male and female offspring at adulthood (P60). Our RNA-seq data analysis suggests that the changes in RNA expression in response to FAE are marked sex-specific. We show that the genes Muc3a, Pttg1, Rec8, Clcnka, Capn11, and pnp2 exhibit significantly higher expression in the male offspring than in the female offspring at P60. FAE female mouse brain sequencing data also show an increased expression of Eno1, Tpm3, and Pcdhb2 compared to male offspring. We performed a pathway analysis using a commercial software package (Ingenuity Pathway Analysis). We found that the sex-specific top regulator genes (Rictor, Gaba, Fmri, Mlxipl) are highly associated with eIF2 (translation initiation), synaptogenesis (the formation of synapses between neurons in the nervous system), sirtuin (metabolic regulation), and estrogen receptor (involved in obesity, aging, and cancer) signaling. Taken together, our transcriptomic results demonstrate that FAE differentially alters RNA expression in the adult brain in a sex-specific manner.

## 1. Introduction

Alcohol consumption during pregnancy causes several neurobehavioral abnormalities in children, commonly referred to as fetal alcohol spectrum disorders (FASDs) [1,2]. Deficits in social interaction and reduced social memory, especially among males, were demonstrated in models of FASDs [3,4]. Worldwide, approximately 10% of women in the general population use alcohol during pregnancy, though the European region has the highest consumption rate (25%) [5,6]. The COVID-19 pandemic exacerbated this risk, and strategies are urgently needed to prevent FASDs from becoming COVID-19 collateral damage [7].

Alcohol is a teratogenic agent that crosses the placental barrier and affects brain structure and function [8]. Fetal alcohol exposure (FAE) can significantly affect fetal development, including changes in brain gene expression [9]. These changes can have lasting effects on the reduction in brain volume, leading to a range of cognitive and behavioral problems in children exposed to alcohol prenatally [10,11]. FAE also affects CNS-associated problems, and regulates candidate gene expressions related to known neurological issues, including Parkinson’s disease, Alzheimer’s disease, Huntington’s disease, and multiple sclerosis. Evidence suggests that FAE regulates oxidative phosphorylation in these offspring [12,13,14,15].

Previously, we have shown that FAE induces sex-specific changes, including inflammatory gene expression changes in the adult brain cortex [4]. The influence of ethanol consumption during pregnancy affects epigenome and embryonic development, significantly impacting adulthood [16]. Other FAE studies also show that brain candidate genes have been associated with FAE outcomes [17], but a comprehensive sex-specific whole-transcriptomic analysis in the cerebral cortex has not been conducted in a FAE model. The availability of high-throughput transcriptomic data and their computational analysis methods enable genome-wide transcriptome profiling, and provide an opportunity to identify biomarkers to facilitate the diagnosis, prognosis, and treatment of patients. However, access to brain tissue in living patients is not possible. Mouse models provide relevant information for some brain diseases. The primary hippocampal cell culture system and drosophila model have also been used for FAE-associated gene expression analyses [18,19].

In this study, we used an established model and performed a FAE-induced transcriptomic analysis in the cerebral cortex of male and female offspring and determined the sex and treatment effects. We also evaluated the gene and pathways associated with FAE outcomes.

## 2. Materials and Methods

### 2.1. Fetal Alcohol Exposure

Female C57BL/6 mice were purchased from Jackson Laboratories and bred in our colony. All animal experiments were performed according to the protocol approved by the University of Tennessee Health Science Center (UTHSC) Institutional Animal Care and Use Committee. Mice were housed in groups of 2–5 per cage, segregated by sex, in a room on a 12 h/12 h light/dark cycle (lights on at 8:00 AM, off at 8:00 PM) maintained at 22 ± two °C. Pregnant mice were fed with or without EtOH (0% 2d, 1% 2d, 2% 2d, 4% one week, and 5% one week) in a Lieber–DeCarli liquid diet. The control group was pair-fed with an isocaloric diet. Pups were separated based on their sex on postnatal day 21. The whole-transcriptomic analysis was performed at postnatal day 60.

### 2.2. Plasma EtOH Concentration Analysis

Plasma samples were analyzed using the Colorimetric Alcohol Assay Kit (STA-620; Cell Biolabs, Inc, San Diego, CA, USA) according to the vendor’s instructions. Briefly, the stock of EtOH (200mM) was diluted in a 1X assay buffer to produce standards in the concentration range of 0–200 μM. Next, all samples were diluted in the 1X assay buffer (1:80). Following this, 10 μL of the diluted EtOH standards or samples was added to a 96-well microtiter plate. The reaction mixture (90 μL) containing the assay buffer, enzyme mixture, and colorimetric probe was added to each well in a microtiter plate. The plate was incubated at 37 °C for 30 minutes and read at 570 nm. The EtOH concentration of samples was calculated.

### 2.3. RNA Isolation and Next-Generation Sequencing

On postnatal day 60, mice were sacrificed under anesthesia (Isothesia). The cerebral cortex was dissected from the whole brain. Samples were transferred to individual tubes containing RNAlater and stored at −80 °C. Total RNA was isolated using a commercial RNA extraction kit, PureLink RNA Mini Kit (Invitrogen). RNA samples were sent to GENEWIZ for library preparation and next-generation sequencing (NGS) using the Illumina platform, and we collected the transcriptomic files.

### 2.4. RNA-Seq Analysis

We used the Multiqc tool [20] to conduct the quality control checks on the FASTQ files. The Ensembl genome assembly GRCm39 was used as a reference, and the gene transfer format (GTF) GRCm39.104 was used for the transcript/gene annotations. The reads were mapped to the Mus musculus genome assembly GRCm39 using a STAR v2.7 aligner [21]. We used the gene count option in RSEM STAR to estimate the read counts per gene [22]. Differential gene expression was performed using the Bioconductor tool, limma [23], package in R by first normalizing the read counts with the voom [24] function in limma and removing the genes that did not contain expression values in more than 25% of the samples [23]. The significant genes in EtOH-fed males compared to the EtOH-fed female were identified with a cutoff log2-fold change > +/− 1 and the Benjamini–Hochberg (BH)-adjusted *p*-value < 0.05.

### 2.5. GSEA Pathway Analysis

For the pathway analysis, we used the gene set enrichment analysis (GSEA) [25] in the Bioconductor tool clusterProfiler [26] by using the fgsea tool in the background. For the pathway analysis, we used minimum and maximum gene sizes for each pathway, 5 and 100, respectively, and the FDR adjusted the *p*-value of 0.05. Entrez gene ids were utilized with a BH multiple adjustment threshold of 0.05, with a minimum and maximum of 5 and 500 genes for each pathway. KEGG and Reactome were used for the pathway enrichment analysis.

### 2.6. Ingenuity Pathway Analysis

IPA^®^ (https://digitalinsights.qiagen.com/products-overview/discovery-insights-portfolio/analysis-and-visualization/qiagen-ipa/, accessed on 10 October 2022) was used for the analysis, integration, and interpretation of data derived from the RNA-seq experiments. IPA mapped genes with Entrez IDs to the known pathways in the IPA database. Genes such as long intergenic nonprotein coding RNAs (lncRNA) were excluded, as the IPA did not recognize them. Significant biological pathways were identified with the plausible molecular mechanism of differentially expressed genes. We also analyzed the molecular and cellular function of diseases and disorders using the IPA tool.

### 2.7. Statistical Analysis

We conducted all statistical data analyses using R-4.2.2, and the plot was generated using the R package ggplot2 package. We used the *p*-value and adjusted it to 0.05 until and unless a specific specified value was used.

## 3. Results

There were no noticeable morphological changes in the neonates of different groups. The litter size and birth weight of offspring of EtOH-fed dams were significantly lower compared to pair-fed dams. We observed a comparable increased plasma alcohol level (72.23 ± 4.70, mg/dL) in the pregnant mice after 20 days of EtOH feeding. FAE is known to affect specific candidate genes in the mouse brain. We evaluated the sex-specific whole-transcript levels in the cerebral cortex with RNA-Seq. We observed significant sex-specific changes in the FAE offspring. Out of 15227 genes, 224 genes were upregulated in the EtOH-fed male offspring compared to the pair-fed male offspring. A total of 271 genes was downregulated in the EtOH-fed male offspring compared to the pair-fed male offspring (Appendix A). The top genes are shown in a volcano plot (Appendix A).

Similarly, 37 genes were upregulated in the EtOH-fed female offspring compared to the pair-fed female offspring. Fifty-one genes were downregulated in the EtOH-fed female offspring compared to the pair-fed female offspring (Appendix A). The top genes are shown in a volcano plot (Appendix A). Interestingly, the differential expressions in EtOH-fed females and EtOH-fed males revealed that 1362 genes were differentially expressed. Of these, 765 genes were differentially upregulated and 597 downregulated (Figure 1A). Details about the genes are included as Appendix A.

KEGG and IPA pathway analyses of differentially expressed genes in EtOH-fed males compared to their counterpart females suggested that neurological-disorder-related pathways were enriched (Appendix A). A list of the top enriched KEGG pathways (Figure 1B) and GSEA plots of known neurological-disorder-related pathways are available in Figure 1C–F. We further analyzed differentially expressed genes using IPA and reconfirmed the enrichment of neurological and psychological disorders (Table 1, Figure 2, and Appendix A). A positive net enrichment score (NES) in the GSEA-based pathway analysis (Figure 1A,C–F) and Z-score in the IPA analysis (Figure 2 and Appendix A) suggest that neurological- and psychological-disorder-related pathways were upregulated in male offspring compared to female offspring.

Using the IPA tool, we determined the known disease signaling pathways that were affected as a significant possible consequence of ethanol feeding during gestation. Our data suggest that differentially expressed transcripts in the male offspring were associated with neurological disorders, including Alzheimer’s disease, Parkinson’s disease, and Huntington’s disease. On the other hand, in females, upregulated transcripts were related to oxidative phosphorylation and ATP production (Figure 2 and Table 1). Details about IPA pathways and diseases and disorders are available as Appendix A.

We visualized the sex and treatment effect with a heatmap. Thirty of the most variable genes were used (based on MAD) for the visualization and clustering analysis. The clustering analysis suggested a clear difference in samples, with no mixing of samples. Individual transcript levels in the FAE cerebral brain cortex clearly distinguished the sex and treatment effect of ethanol feeding during gestation (Figure 3).

We summarized sex and ethanol treatment effect transcriptome data using a Venn diagram. We observed that 229 gene expressions (82.1%) were altered in the male offspring, and 32 gene (11.5%) expressions were altered in the female offspring. Eighteen (6.5%) common transcript expressions were changed in both male and female offspring. Common differentially expressed genes (18) were also included in the plot (Figure 4). A list of genes that were exclusively expressed in the male and female offspring is available as Appendix A.

## 4. Discussion

FASDs are a major concern in neonatal clinics, and the mechanisms associated with the pathogenesis of these disorder are not well understood. Maternal alcohol consumption during pregnancy affects the fetus’s development, but the know-how of its regulation for offspring behavior and brain functions is unknown. FAE can lead to a range of negative consequences for the developing brain. The whole-transcriptome analysis is a powerful tool that can help researchers understand the complex molecular changes that occur in the brain due to fetal alcohol exposure [17]. Prenatal alcohol exposure also contributes to adverse pregnancy outcomes by altering fetal vascular dynamics and the placental transcriptome [27]. Ex vivo primary hippocampal cell culture systems have also validated FAE-associated gene expression changes. In both males and females, hippocampal transcript levels differed by prenatal treatment, with sex differences observed in the expression of the insulin-like growth factor 2 and insulin receptors. The effect of prenatal EtOH on the hippocampal expression of the insulin pathway genes was parallel in the in vivo and the ex vivo conditions [18]. By analyzing the entire transcriptome, or the complete set of genetic instructions within a cell, researchers can identify specific genes and pathways that may be impacted by alcohol exposure, which could help identify potential therapeutic targets and interventions. Overall, the whole-transcriptome analysis could significantly advance our understanding of the effects of FAE on the developing brain, and may help identify new approaches to prevent or mitigate these adverse outcomes [19,28]. Our study investigated the impact of the FAE-induced whole transcriptome and its association with known signaling pathways in the cerebral cortex of offspring.

The results indicated that ethanol feeding in mice during gestation significantly induced sex-specific changes in the cerebral cortex of the offspring. In the previous study, we showed that fetal alcohol induced sex-specific social interaction deficits and inflammatory gene (IL-1β, IL-6, and TNF-α) expression. We also demonstrated that FAE induced some known developmental genes, including Mecp2, Ube3a, and Gabr3b [4,29]. This previous study also revealed that FAE interfered with social interaction in adult males, but not females [4].

Earlier, we showed that FAE reduces the litter size and lowers the birth weight in both mice and rats [4,29]. We reproduced similar observations in this study as well. Our current study showed that FAE induced sex-specific changes in the genes both in males and females. The male Muc3a, Pttg1, Rec8, Clcnka, Capn11, and pnp2 gene expression levels significantly increased. Muc3a is a known protein-coding gene associated with inflammatory bowel disease and several cancers, but its role in the FAE brain has not yet been studied [30,31]. Pttg1 (pituitary tumor transforming gene 1) showed significant changes in the male offspring [32]. Rec8, a meiosis-specific component of cohesin, plays a central role in the recombination of homologous chromosomes during meiosis [33]. Clcnka (chloride voltage-gated channel Ka) is associated with Bartter syndrome. Bartter’s syndrome describes a group of disorders unified by the autosomal recessive transmission of altered salt reabsorption. Patients with a neonatal form of Bartter syndrome typically present with premature birth associated with polyhydramnios and a low birth weight, and may develop life-threatening dehydration in the neonatal period [34,35].

A human testis-specific calpain 11 (CAPN11) gene was identified with motifs for protease activity [36], but its function in the FAE brain was not studied. Calcium-regulated nonlysosomal thiol protease catalyzes the limited proteolysis of substrates involved in cytoskeletal remodeling, signal transduction, and heart defects in children [37,38].

Bartter syndrome refers to a group of disorders that are unified by the autosomal recessive transmission of impaired salt reabsorption in the thick ascending loop of Henle, with pronounced salt wasting, hypokalemic metabolic alkalosis, and hypercalciuria. Clinical disease results from defective renal sodium chloride reabsorption in the Henle loop’s thick ascending limb (TAL) showed that 30% of filtered salt was usually reabsorbed [39]. Patients with antenatal (or neonatal) forms of Bartter syndrome typically present with premature birth associated with polyhydramnios and a low birth weight. They may develop life-threatening dehydration in the neonatal period. Patients with classic Bartter syndrome present later in life, and may be sporadically asymptomatic or mildly symptomatic [40]. The PNP gene provides instructions to manufacture an enzyme known as nucleoside purine phosphorylase. More than 35 PNP gene mutations have been identified in individuals with a purine nucleoside phosphorylase deficiency, an immune system disorder in which the body cannot fight foreign invaders such as bacteria and viruses. However, the pnd2 role in FAE is still unknown [41].

Eno1, Tpm3, and Pcdhb2 gene expression levels were significantly increased in the female FAE offspring. The overexpression of ENO1 is linked to glioma progression, and its knockdown resulted in the suppression of cell growth, migration, and invasion progression by inactivating the PI3K/Akt pathway in glioma cells [42]. The remodeling of the actin cytoskeleton is controlled by tropomyosins, a family of actin-associated proteins that define distinct actin filament populations. The previous study showed that TPM3 and TPM4 gene products localized to the postsynaptic region in mouse hippocampal neurons. TPM3 is one of the genes whose effect segregates to the postsynaptic part of the central nervous system synapses [43]. Its role in the FAE model has not been investigated, where cell–cell interaction was compromised during brain development [44]. The Pcdhb2 gene is a member of the protocadherin beta gene cluster. Their specific functions with fetal alcohol are unknown, but they most likely play a critical role in establishing and operating cell–cell neural connections [45]. Together, these results suggest that fetal alcohol exposure induces sex-specific changes in the cerebral transcriptome. The heatmap and trait association analysis revealed a differential level of transcripts and a specific association between the groups. The Venn diagram demonstrated the differential expression of genes in the FAE male and female offspring. These differential sets of genes were associated with known signaling pathways, which elaborated the importance of FAE-related sex-specific outcomes that can regulate during the gestational and postnatal periods. The previous study showed that some autistic candidate genes were associated with FAE and had a significant sex-specific impact on fetal development.

This study showed that EtOH feeding during gestation induces sex-specific changes in the critical brain transcripts. We speculated that these changes in transcriptome may involve an altered sex-specific behavior and other outcomes, including developmental abnormalities in the FAE offspring.

## 5. Conclusions

Fetal alcohol exposure (FAE) induced sex-specific changes in the brain cortex. FAE altered critical signaling pathways in the brain cortex of male and female offspring.

## Figures and Tables

**Figure 1 cells-12-00328-f001:**
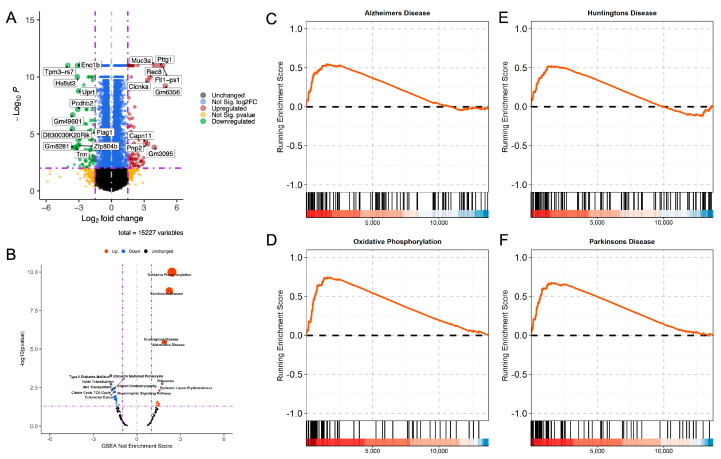
Gene expression changes and associated disease pathways in the FAE offspring. Volcano plot for the differentially expressed genes. Genes in red and green colors were highly upregulated and downregulated in EtOH-fed males compared to EtOH-fed females. Vertical and horizontal dot lines represent a cutoff point for log-fold change *p*-value, respectively. Genes in orange were differentially expressed, but the *p*-value was not significant. Similarly, genes in blue had a meaningful *p*-value, but the log2-fold change was < 1.5. The top differentially expressed gene names are presented in their respective volcano plots: (**A**) GSEA Volcano plot; EtOH-fed male compared to EtOH-fed female, (**B**) GSEA plot shows upregulated genes in FAE male offspring were highly associated with Alzheimer’s Disease (**C**), Oxidative Phosphorylation (**D**), Huntington’s Disease (**E**), and Parkinson’s Disease (**F**).

**Figure 2 cells-12-00328-f002:**
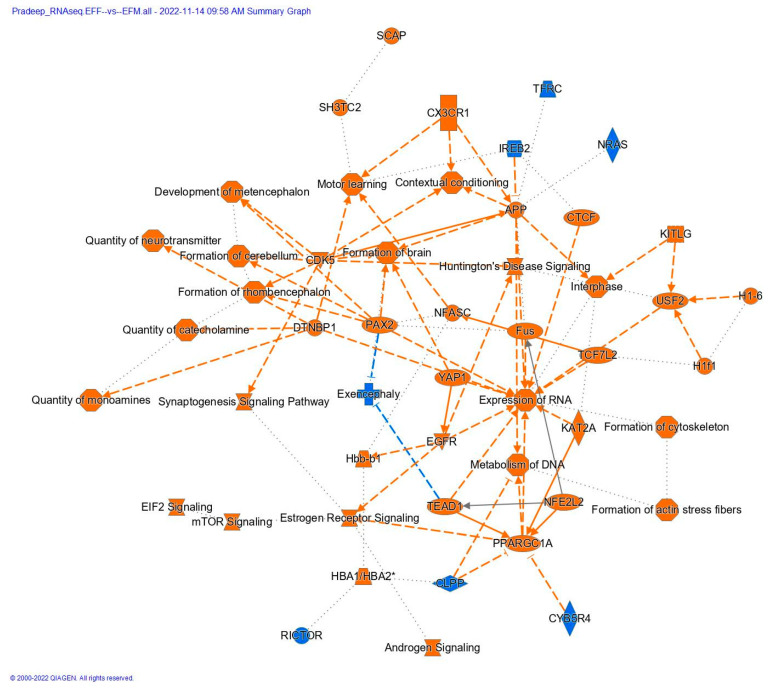
Ingenuity pathway analysis of differentially expressed EtOH-fed males compared to EtOH-fed females. IPA analysis suggested a lot of neurological degenerative diseases and neurotransmitter-related pathways were upregulated in males compared to females. IPA analysis also indicated that a lot of critical transcription factors, which were highly interconnected, were also upregulated in males.

**Figure 3 cells-12-00328-f003:**
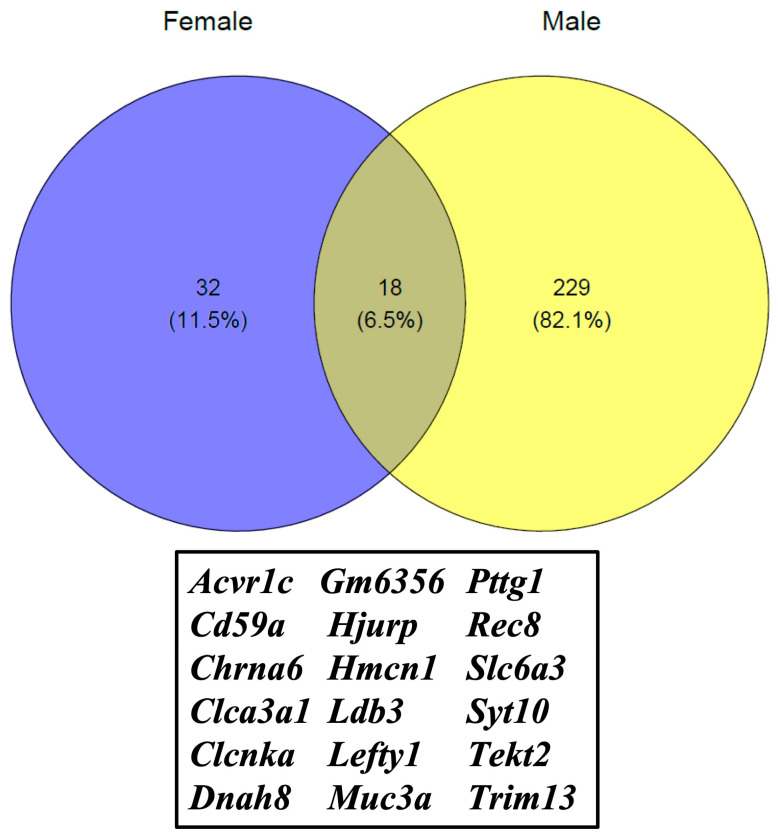
Venn diagram for common or exclusively expressed genes in males and females, respectively. In this analysis, we compared genes differentially expressed in EtOH-fed male/female samples to corresponding control (PF) male/female samples.

**Figure 4 cells-12-00328-f004:**
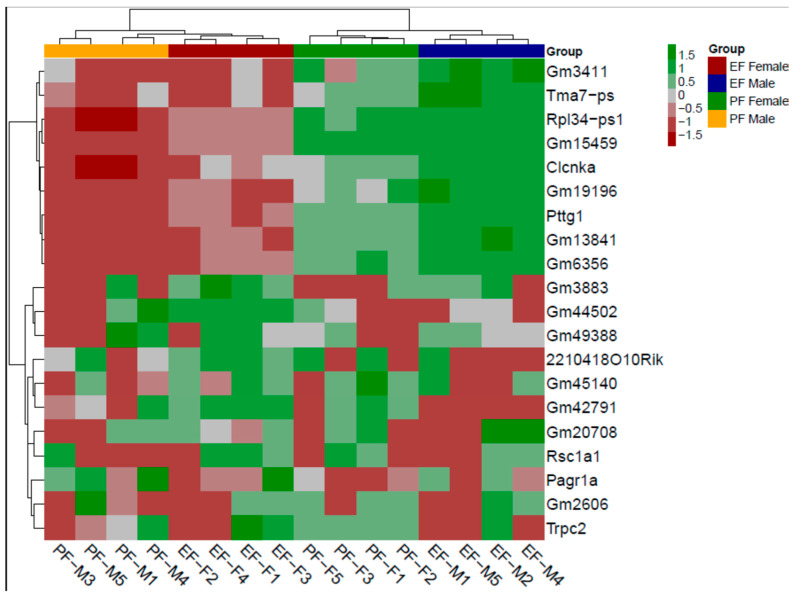
Unsupervised hierarchical clustering of most variable 30 genes in samples (columns) based on median absolute deviation (MAD). These 30 genes had the most variability across all models. The green color in the heatmap indicates a higher expression level, and red indicates a low expression level. In groups, each color represents the respective groups, as explained in the legend (M—male; F—female).

**Table 1 cells-12-00328-t001:** Top five genes in terms of sex-specific changes in gene expression (EtOH-fed female vs. EtOH-fed male).

Term	*p*-Value
**IPA Pathways**	
EIF2 signaling	2.75 × 10^−23^
Huntington’s disease signaling	1.98 × 10^−20^
Synaptogenesis signaling pathway	2.93 × 10^−16^
Sirtuin signaling pathway	9.75 × 10^−16^
Estrogen receptor signaling	2.02 × 10^−14^
**KEGG Pathways**	
Oxidative phosphorylation	1.86 × 10^−08^
Parkinson’s disease	1.57 × 10^−07^
Alzheimer’s disease	0.000163
Huntington’s disease	0.000163
Type II diabetes mellitus	0.019792
**IPA Diseases and Disorders**	
Organismal injury and abnormalities	1.07 × 10^−03^
Neurological diseases	1.07 × 10^−03^
Hereditary disorders	6.39 × 10^−04^
Psychological disorders	6.39 × 10^−04^
Skeletal and muscular disorders	6.39 × 10^−04^
**IPA Molecular and Cellular Functions**	
Cellular development	1.05 × 10^−03^
Cellular growth and proliferation	1.05 × 10^−03^
Cell morphology	1.07 × 10^−03^
Cellular assembly and organization	1.08 × 10^−03^
Cellular function and maintenance	1.02 × 10^−03^

## Data Availability

Not applicable.

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
