# Peer review of "Sex-Specific Whole-Transcriptome Analysis in the Cerebral Cortex of FAE Offspring"

_cells, 2023, doi:10.3390/cells12020328_

Round 1

Reviewer 1 Report

Abstract:ok

Introduction: ok

Materials and methods: It is stated that the "cerebral cortex was dissected from the whole brain" Did These ´specimen contain neocortex only? 

Results: ok

Discussion: It is stated that expression levels of genes causing inflammatory bowel disese, pituitary tumors and Bartter Syndrome are elevated in male offspríngs. However, to date there is very little evidence that this happens also in humans- how can that be explained? 

Conclusion: ok

Author Response

Thanks a lot for the reviewers' comments. Now we have answered all the concerns raised by learned reviewers and included in the revised manuscript.

Materials and methods: It is stated that the "cerebral cortex was dissected from the whole brain" Did These ´specimen contain neocortex only? 

Yes, we have used cortex including neocortex.

Discussion: It is stated that expression levels of genes causing inflammatory bowel disease, pituitary tumors and Bartter Syndrome are elevated in male offspríngs. However, to date there is very little evidence that this happens also in humans- how can that be explained? 

This is a first comprehensive RNA seq analysis in the cerebral cortex of FAE offspring. I agree with the learned reviewer that we don't have additional information related to these genes particularly in FAE offspring. In future we wish to further explore these candidate genes in the FAE model.

Reviewer 2 Report

The submitted manuscripts seeks to investigate sex specific changes in RNA profiles of FAE offspring in mice.   The authors report several altered levels that are differentiated by sex and attempt to use the levels to provide rationale for altered birthweights and growth.   

The paper itself is quite shallow in terms of the work done and several key components are missing.  These are detailed here and need to be corrected.

1. The strain of the mice is not given.  This is critical and the authors should also speculate on how mouse strain may impact their data.

2.  Ethanol is said to be administered via liquid diet.  Was water available?

3. It appears no iso-caloric diet was used as a control.  Why was this not used?   In addition, this needs to be discussed as a limitation of the study.

4. How did the authors handle pups?   Where they culled by a number per sex?  

5.  Was blood ethanol levels determined?  If not, why’d  not?

6.  The introduction needs much work in terms of behavioral measures impacted by FAE and the discussion should address gene changes that may impact the previous behavioral changes reported in the literature.

Author Response

The submitted manuscript seeks to investigate sex specific changes in RNA profiles of FAE offspring in mice.   The authors report several altered levels that are differentiated by sex and attempt to use the levels to provide rationale for altered birth weights and growth.   

The paper itself is quite shallow in terms of the work done and several key components are missing.  These are detailed here and need to be corrected.

Thanks a lot for the reviewers' comments. Now we have answered all the concerns raised by learned reviewers and included in the revised manuscript.

1. The strain of the mice is not given.  This is critical and the authors should also speculate on how mouse strain may impact their data.

We have used C57BL/6. Now it's been included in revised manuscript

2.  Ethanol is said to be administered via liquid diet.  Was water available?

We have used a special ethanol liquid diet with water throughout gestation. It is a standard diet used in alcohol research.

3. It appears no iso-caloric diet was used as a control.  Why was this not used?   In addition, this needs to be discussed as a limitation of the study.

We have used a standard etanol diet and corresponding controlled diet (maltodextrin to match ethanol calorie) which is well established in  alcohol research.

4. How did the authors handle pups?   Were they culled by a number per sex?  

Pups were separated at postnatal day 21 based on their sex. Now it's been clarified in the revised manuscript. 

5.  Was blood ethanol levels determined?  If not, why’d  not?

Blood alcohol levels were measured (73.4+ 10.3 mg/dl) in this study and included in our previous fetal alcohol study (PMID: 30312858).

6.  The introduction needs much work in terms of behavioral measures impacted by FAE and the discussion should address gene changes that may impact the previous behavioral changes reported in the literature.

Agree with the learned reviewer and it's been extensively revised. 

Shukla PK, Meena AS, Rao R, Rao R. Deletion of TLR-4 attenuates fetal alcohol exposure-induced gene expression and social interaction deficits. Alcohol. 2018 Dec;73:73-78. doi: 10.1016/j.alcohol.2018.04.004. Epub 2018 Apr 18. PMID: 30312858; PMCID: PMC6193854.

Round 2

Reviewer 2 Report

The authors have addressed a portion of my initial concerns and simply ignored others.  For example, they state the measured BAC in a previous paper but that needs to be stated in this paper.  Also, are these the same animals from the previous paper? if not, then the authors don't really know how much ethanol is on board.  They would need to state that and then cite the original paper.  Second, the I asked the authors to really revise the introduction to put into context the work here in terms of the well known effects of FAE.  The authors state they have done so.  In the revised manuscript with track changes there are no changes in the introduction and a poorly written paragraph in the discussion.  Therefore this still needs to be done and is actually quite frustrating that they authors decided to ignore the comment.

Author Response

R2-Point-by-point response

Thanks for your time and effort. Just want to bring in your kind knowledge that our first revised submission did not go through properly. We have added 8 new references. Now it's been taken care of. I am sorry for any inconvenience.   

Looking forward to hearing from you.

Pradeep

Point-by-point response

The authors have addressed a portion of my initial concerns and simply ignored others.  For example, they state the measured BAC in a previous paper but that needs to be stated in this paper.  Also, are these the same animals from the previous paper? If not, then the authors don't really know how much ethanol is on board.  They would need to state that and then cite the original paper. 

This is a separate new study. BAC data has been included in the revised version (Method page#2, line#71-81; Resultpage#3, line#182-183). 

Second, the I asked the authors to really revise the introduction to put into context the work here in terms of the well known effects of FAE.  The authors state they have done so.  In the revised manuscript with track changes there are no changes in the introduction and a poorly written paragraph in the discussion.  Therefore this still needs to be done and is actually quite frustrating that the authors decided to ignore the comment.

Sorry for the trouble, introduction and discussion have been revised substantially, (Introduction page#1, line#29-58, Discussion, page#13, line#330-434). It was an inadvertent mistake during the first revision.

Round 3

Reviewer 2 Report

Nil.